# COVID-19, Anti-Intellectualism, and Health Communication: Assessing the Chinese Social Media Platform Sina Weibo

**DOI:** 10.3390/healthcare11010121

**Published:** 2022-12-30

**Authors:** Shouyun Guo, Teng Lin, Nadeem Akhtar, Juana Du

**Affiliations:** 1School of Urban Culture, South China Normal University, Guangzhou 510631, China; 2Pakistan Center, School of Ethnology, North Minzu University, Yinchuan 750021, China; 3School of Communication and Culture, Royal Roads University, Victoria, BC V9B5Y2, Canada

**Keywords:** anti-intellectualism, China, health information, new media, textual analysis

## Abstract

In the digital media era, new media platforms have become the main channels for transmitting medical and health information in China. However, anti-intellectualism limits the effectiveness of disseminating health information. Therefore, in China, the government and health departments have made efforts to determine how to control anti-intellectualism to effectively disseminate medical and health information, given the situation of a global pandemic and its counter-measures. Against this backdrop, this study applied textual analysis to explore the manifestations of anti-intellectualism in network platforms. The key findings indicate that the irrational behavior of anti-intellectuals is manifested in emotional dominance, abusive behavior, overconfidence and trusting rumors. Based on these results, the authors propose some measures to balance the relationship between anti-intellectualism and health communication. The findings of the study have significant implications for improving the effectiveness of health communication in China.

## 1. Introduction

Since the COVID-19 outbreak in January 2020, the Chinese government has used a variety of measures to control the spread of it, with the dissemination of public health and health science information being one of the key measures [1,2]. The communication of public health and health science information is one of the elements of health communication. With COVID-19 raging, the importance of health communication is becoming increasingly evident. The communication of health information aims to disseminate medical research results to the audience to impact their attitudes and behaviors, thereby reducing the transmission and mortality rate of diseases and effectively improving the quality of life and health of the public [3]. In the age of digital media, new media platforms have been regarded as the main channels for communicating medical and health information due to their fast speed and high information storage capacity. Previous studies have also shown the growing importance of new media platforms in disseminating and communicating medical and health information [4,5]. By disseminating health information on new media platforms, medical experts or professional organizations can enable a broader audience to quickly learn practical ways to prevent and control epidemics and take an active approach. As a result, the risk of COVID-19 to the audience is reduced and the consumption of medical resources decreases.

However, while new media platforms have significantly contributed to health communication, they have also led to the proliferation of anti-intellectualism. In Hofstadter’s view, anti-intellectualism includes forms of distrust and even attacks on experts and intellectuals as well as the social elite. With the development of new media platforms, anti-intellectualism has begun to emerge across new media platforms. Previous research has shown that social media platforms such as Twitter and Facebook are common places where malicious attacks on experts and academics exist [6,7,8]. In China, social media platforms are also home to many anti-intellectual behaviors [9,10]. Therefore, although new media platforms can improve the efficiency of health communication, they also contribute to the emergence of anti-intellectualism. Previous research has shown that the phenomenon of anti-intellectualism is a barrier to health communication and scientific communication. Some studies have shown that anti-intellectualism leads to phenomena such as public rejection of the scientific consensus and endorsement of misinformed health-related and scientific beliefs [11]. It has also been suggested that anti-intellectualism is a fundamental challenge to increasing the public’s adherence to expert advice [12]. In the context of COVID-19 prevention and control activities, anti-intellectualism has influenced the perceptions of the public regarding information passed on by experts, thus affecting efforts to control COVID-19. For example, recent studies have demonstrated that anti-intellectualism can lead some members of the public to believe misinformation about the vaccine [12,13,14]. Anti-intellectualism in health communication is manifested as the blind and emotional resistance of anti-intellectuals to the health information disseminated by medical experts or professional medical institutions. The texts produced by this resistance are circulated on new media platforms, making some viewers doubt the health information published by medical experts and weakening the impact of disseminating medical and health-related information. These emotional expressions of anti-intellectualism are not only found in the area of health communication, but are also present in events that have a significant effect, such as disasters [15,16] or newly introduced government policies [17]. These findings contribute to current literature on anti-intellectualism, which is an emerging research topic with a very limited number of empirical; further, this research identifies several manifestations of anti-intellectualism by analyzing health messages distributed on Sina Weibo in the context of China, and provides empirical data to study the characteristics of anti-intellectualism in online health communication. Last but not least, the research findings on balancing measures help to advance understandings of anti-intellectualism and expand the framework of anti-intellectualism to incorporate strategies improving effective health communication online.

These research findings have several practical implications for health communication authorities and professionals. First, they raise public awareness of the importance of anti-intellectualism in health information dissemination and its impact on public perceptions of health information. Second, they suggest health communication authorities develop active strategies to improve the efficiency of health information dissemination on new media platforms. Last but not least, they further suggest for health communication professionals to consider the manifestations of anti-intellectualism on online platforms in the context of China, and design culturally appropriate communication methods to support effective health information dissemination.

This article is structured as follows. First, we reviewed previous relevant studies to develop a theoretical framework. Next, we collected and analyzed comments from anti-intellectuals on the messages delivered by experts in the health field and discussed them. In the discussion section, we also present suggestions for balancing the effects of anti-intellectualism on the effectiveness of health communication via the internet. Lastly, we conclude with the theoretical and practical implications and suggest directions for further research.

## 2. Theoretical Framework

### 2.1. Anti-Intellectualism

The concept of anti-intellectualism has raised public awareness since 1960s. According to Hofstadter (1966), anti-intellectualism is an attitude that takes the form of dislike and distrust of rational life and those who are its representatives, and a consistent tendency to devalue this life [18]. In other words, anti-intellectualism involves two interrelated components: hatred and distrust of the intellect and the knowledge it extends and rejection and denial of the intellectuals who represent it and the social elite who represent them [19]. Based on the definition of Hofstadter, subsequent studies have further expanded the definition of anti-intellectualism. Rigney (1991) further categorized it as: (a) religious anti-rationalism, (b) populist anti-elitism that abhors the intellect and (c) unreflective instrumental rationalism [20]. Eigenberger and Sealander (2001) suggested that anti-intellectualism devalues activities that require analytical and critical thinking in favor of those that promote practicality, material trade and efficiency [21]. Finley (2009) described anti-intellectualism as a nihilistic tendency characterized by an abhorrence of excellence and the perception of intellectual pursuits as a sign of a privileged class disconnected from everyday reality [22].

In China, anti-intellectualism has now appeared to be a widespread attitude in the dissemination of information on the internet [9,10]. The characteristics of anti-intellectualism on the Chinese internet are similar to those in other countries. Previous studies have identified that anti-intellectualism in mass communication is mainly characterized by irrational behavior, anti-elitist behavior and overly utilitarian behavior [23]. Some studies have also shown that anti-intellectualism on Chinese online platforms shows the characteristics of promoting extremism against rationality [24]. The main features of anti-intellectualism and its manifestations in health communication are summarized and discussed in the following section.

### 2.2. Irrational Behavior

Irrational behavior is considered a manifestation of anti-intellectualism because several studies have reported that anti-intellectualism leads individuals to lose their intellectuality and abandon critical thinking and hypothetical inquiry [21,25]. Thus, members of the public with anti-intellectual attitudes are characterized by how, instead of appraising information from experts through critical thinking, they selectively receive information based on their own emotions, experiences and preferences [26,27,28]. The selective reception of information by audiences often leads to irrational behavior. Irrational behavior occurs particularly in the face of sudden and devastating situations, such as the “salt panic” in China after the 2011 Tohuku tsunami, when rumors circulated over the internet and text messages, caused “panicbuying” of salt, and the rush to buy Shuanghuanglian (an unproven traditional Chinese medicine against COVID-19) after the COVID-19 outbreak [29,30]. In these incidents, the advice of experts was ignored or discarded.

Several studies have analyzed the characteristics of members of the general public with irrational behavior. For example, in a study conducted in Jordan using questionnaires, Sallam et al. (2020) found that irrational people had higher levels of anxiety, which prompted them to be more likely to believe rumors or conspiracy theories [31]. In a study conducted in China, also using a questionnaire, researchers found that people with irrational behavior tended to have negative attitudes toward the news they received [32]. Miller et al.’s (2016) study, after comparing and analyzing the data from two groups, concluded that people with irrational behavior tended to have confidence in their knowledge in a particular area while lacking trust in authority figures [26]. In general, previous studies have focused on the mental states or attitudes of irrational people rather than directly observing their behavior. Again, most studies have focus on irrational behavior, not anti-intellectual behavior. The relationship between these is that there are many manifestations of irrational behavior, and irrational anti-intellectual behavior is only one of them. No studies specifically focusing on irrational anti-intellectual behavior were found at present.

### 2.3. Opposition to Intellectuals and Experts

Blind opposition to expert scholars and elites is one of the manifestations of anti-intellectualism. Hofstadter (1966) defined anti-intellectualism by emphasizing the component of opposition: anti-elitism, which indicates distrust and dislike of experts purporting to have more knowledge on a subject [18]. A number of studies have reported on the causes of this phenomenon. Hofstadter (1966) suggested that experts could be considered dangerous because they occupy the halls of power and claim to know how citizens should better manage their lives [18]. Some studies have also suggested that the cause of this phenomenon could be that people adopt information (including misinformation) that fits their prior political, group-based or attitudinal preferences and reject information that goes against their prior beliefs [26,28,33]. It has also been argued that citizens’ reasons for rejecting expert advice often focus on ideologically motivated reasoning [34,35]. Alternatively, citizens may be skeptical of the knowledge gained because they see it as a tool used by the social elite to exploit the masses [36]. Furthermore, in any pro-democracy society, people expect someone to doubt a well-educated member of the elite’s quality of knowledge or talent [20].

Public opposition to experts has become commonplace in the field of health communication on Chinese internet platforms. Some studies have suggested that experts in China are facing a crisis of trust. A study that surveyed users from across China found that 37% of the respondents did not trust experts, mainly due to the public’s perception that experts represent commercial organizations, as well as the inability to understand the information they disseminate [37]. Another study, also using a questionnaire, surveyed the residents of 36 cities in China and found that 27% of the respondents distrusted experts, mainly due to the unethical behavior of some experts and media reports that misinterpreted the experts’ original intentions. The results of these studies illustrate people’s attitudes toward experts and the reasons for their mistrust [38]. The results of these studies demonstrate people’s attitudes toward experts and the reasons why distrust arises. Surveys using questionnaires can achieve this, but it is difficult to observe specific behaviors. Moreover, during a questionnaire, respondents may give wrong answers because they know they are being surveyed. Thus, a direct study of the behavior of anti-intellectuals may lead to some new conclusions. A previous study analyzed the online audience’s comments on articles published by experts about genetic modification and found that the online community used social rationality with different forms of logic, such as experience, values and professionalism, to confront the genetic modification experts and defeat their scientific reasoning [24]. This approach effectively analyzed the specific behaviors of anti-intellectuals. Therefore, we aimed to use this method to analyze the behavior of anti-intellectuals in the area of health communication.

### 2.4. Unreflective Instrumentalism

As Hofstadter said, “the devaluation of ideas that do not yield immediate benefits is one of the manifestations of anti-intellectualism” [18]. Rigney (1991) summarized this as “unreflective instrumentalism”, which suppresses questions about the ends toward which practical and efficient means are directed [20]. In other words, anti-intellectuals are more concerned with the immediate benefits and discount long-term but not immediately available benefits, even though these long-term benefits may be more beneficial than the immediate benefits. However, in crises, experts or professional bodies need to give appropriate advice and take appropriate measures based on the public’s best interest [39,40]. Thus, unreflective instrumentalism can lead to the rejection of advice from experts when it does not correspond to the public’s need to obtain direct benefits. This unreflective instrumentalism is also evident in China and has led some audiences to pursue profit at the expense of social responsibility [23].

There has been no research on the specific manifestation of unreflective instrumentalism in society. One study has regarded the abandonment of public interest in favor of personal interest as unreflective instrumentalism [41]. However, this is only one possible result of unreflective instrumentalism, not its essence. Therefore, the manifestation of unreflective instrumentalism in society needs to be studied further.

In summary, previous research has identified irrational behavior, opposition to intellectuals and experts, and unreflective instrumentalism as the three main manifestations of anti-intellectualism on Chinese internet platforms, and that these anti-intellectual phenomena can have a negative impact on health communication. Although there has been some research on this topic, there are still some problems. Firstly, most studies have analyzed the mental states, attitudes or opinions of anti-intellectuals, but few have specifically analyzed their behaviors. Therefore, a study of the behavior of anti-intellectuals could provide a new perspective for understanding this group. Second, irrational behavior and opposition to intellectuals and experts have been studied, but few studies have focused on the anti-intellectual behaviors involved. Irrational behaviors can be of many kinds, and irrational anti-intellectual behaviors are only some of them. Opposition to experts is not entirely irrational, but also includes some reasonable questioning that cannot be considered anti-intellectualism. Thirdly, although previous studies have reported that the presence of anti-intellectualism affects the efficiency of health communication, no studies have been found that suggest strategies to balance anti-intellectualism and health communication.

Therefore, this paper will examine the manifestations of anti-intellectualism on Chinese internet platforms and provide empirical data from the cultural context of China. We propose the research questions as following:

How has anti-intellectualism presented in the online dissemination of health information in the context of China?

Two sub-questions are:What are the main characteristics of anti-intellectualism in COVID-19 health information dissemination on online platforms?How have the main characteristics of anti-intellectualism been demonstrated on online platforms in the context of China?

## 3. Methodology

This study took anti-intellectual behavior regarding online health communication as the research object, so the sample of the study was taken from the Chinese internet. On the Internet, anti-intellectual behaviors are mainly reflected in responses to information, including retweeting, liking and commenting on the information. However, retweets and likes contain a limited amount of information, whereas the text of comments can be understood to reveal more information. Therefore, comments from the general public on the health information disseminated by experts were selected as the sample for this study. Comments on the internet are mainly presented in textual form; therefore, we chose textual analysis for studying the sample. The specific sample collection process and analysis process are described below.

### 3.1. Sampling

During data collection, we chose the Sina Weibo platform as the sample source, which ensures the generality and representativeness of the research results. Sina Weibo is currently one of the most popular social media platforms in China, with “511 million active users per month as of September 2020” [42]. In this study, we chose two blog articles and the related public discussions as case studies. The titles of those articles are “High alert! Zhong Nanshan said Omicron is very contagious”, published on the official Weibo page of “People’s Daily” on 23 December 2021, and the article titled “The International Omicron Epidemic is Complicated, and China Responds in an Orderly Way” published by Dr. Wenhong Zhang on 17 December 2021 on his personal Weibo page. Mr. Nanshan Zhong and Dr. Wenhong Zhang are professionals in the field of preventing and controlling infectious diseases. Since the epidemic’s outbreak, they have frequently disseminated information on epidemic-related prevention and control to audiences through new media platforms. We chose those two articles to conduct textual analysis as they appeared as the top two articles in the trending list. This means that they could be picked up by more people who wanted to know more about the “vaccine” issue. Therefore, these two microblog posts are vivid examples of the health information communicated by medical experts through new media platforms. After determining the scope of the sample, we used “Octopus”, which is a web data collector, to collect the comments from both blogs. The principle of Octopus is that by simulating various human Internet behaviors, an automated program is used to replace human browsing and manual copying and pasting of web data, thus automating the collection of the target online data and achieving fully automated data collection by constantly repeating a series of preset actions [43]. We used Weibo’s comment popularity ranking feature to rank the comments on each blog in descending order of “likes” and collected the 800 most liked comments for each post for a total of 1600 comments. Each comment was then manually reviewed to remove comments that were not suitable for sentiment analysis, i.e., those with only emoji and/or symbols. In total, 1540 valid comments were collected.

### 3.2. Textual Analysis

Textual analysis is a qualitative research method that is applied to analyze texts from social media and extract public opinion from them [44]. In this study, we first analyzed the affective tendencies of texts, with the aim of analyzing the emotional reflections of social media users on the advice of experts and academics. Analyzing the affective tendencies of texts can be carried out for words, sentences and paragraphs, with the affective analysis of words being the basis for the other two analyses [45]. Since comments on social media platforms are mainly short sentences and phrases, our analysis also focused on phrases and sentences. At present, research on affective analysis at the word level has mainly been carried out in two ways: the extraction and classification of affective words and the construction of affective dictionaries [46,47]. Some scholars have constructed a theoretical framework for analyzing the affective trends of text, which includes five steps: (a) input document, (b) pre-processing, (c) sentiment classification, (d) distinguishing between positive and negative, and (e) drawing the final conclusions [48,49]. In the affective classification stage, the current mainstream classification methods are machine-learning-based classification and affective-dictionary-based classification. This study used the method of affective-dictionary-based classification to analyze the affective tendencies of the text. The process was as follows: (a) extracting emotional words, negatives, adverbs and conjunctions from the sentences; (b) calculating the emotional weights of the words; (c) considering the effect of conjunctions and the weight of emotional phrases in the sentences; and (d) deriving the emotional tendency of the sentences [45]. Ultimately, we were able to distinguish which texts had negative, positive and neutral affective tendencies. To conduct this step, we used GooSeeker, an analytical software package based on a textual sentiment dictionary to automatically analyze the textual data.

Based on the results obtained from the analysis of affective tendencies, the content characteristics of texts with a negative affective tendency were analyzed further. Textual analysis is a systematic and objective quantitative research method used for studying the characteristics of information [50]. Previous research has suggested that content analysis helps to uncover the hidden content and potential information value of textual information [51,52]. Firstly, we used word frequency analysis to analyze the word usage of these texts. Word frequency analysis is a tool to understand the patterns of word use of text communicators and, in turn, uncover the main ideas present in these texts [53]. A random sample of 100 texts from texts with a tendency towards negative sentiment was used for content analysis.

Secondly, we analyzed the randomly selected texts with negative affective tendencies by coding them sentence by sentence according to their characteristics. These features were then categorized and integrated on the basis of their commonalities to derive the main characteristics of anti-intellectualist textual expressions on Chinese social media platforms. Text coding analysis was applicable to this study because text coding can transform a large number of fragmented, qualitative texts into a systematic, quantitative form, transforming the research object into an easily analyzable dataset that can be further analyzed statistically or extrapolated to infer valuable research findings [54,55].

In the process of text coding analysis, we analyzed each comment with negative affective tendencies, judged its meaning and coded it. To reduce the subjectivity of the judgment process, three researchers coded all the comments simultaneously. If there was a situation where the three researchers had different codes for the same text, the text was discarded. If there was a situation where one researcher had a different coding from the other two, the coding determined by the two researchers was selected. Subsequently, we grouped texts with the same codes in the same first-level category and named this category according to its common characteristics. After that, we constructed three secondary categories based on the three characteristics of anti-intellectualism, namely irrational behavior, opposition to intellectuals and experts, and unreflective instrumentalism. Finally, we assigned all the first-level categories to different second-level categories according to their nature.

We used word frequency analysis and content coding with Nvivo11, a widely used qualitative analytical software package from QSR. Based on this, the theoretical framework of this study was as shown in Figure 1.

## 4. Results

### 4.1. Overall Affective Tendencies

After we analyzed the affective tendencies of the keywords in the collected comments, the results show the audience’s attitudes towards the two tweets (Figure 2). Among the 1540 samples, 448 texts were categorized to have negative sentiments and 410 texts were judged to have positive tendencies, accounting for 29% and 27% of the total number of samples, respectively. In addition, there were 682 texts that were not found to have obvious emotional tendencies, accounting for 44% of the total number of samples. Among them, the texts with negative emotional tendencies indicated that social media users exhibited negative emotions through their texts, including anger, disgust, sadness, fear and questioning the advice given by the experts.

### 4.2. Keyword Visualization

Keyword visualization is used to show the keywords appearing in the text in the form of pictures, and its purpose is to clearly reflect the text’s focus, word usage and logic of expression. The results of the keyword visualization are shown in Figure 3.

Figure 3 shows the 150 most frequently mentioned keywords with negative emotional tendencies. The font size in the figure indicates the frequency, i.e., a larger font indicates a higher frequency. Some of the high-frequency keywords are presented in Table 1. In anti-intellectual discourse, “internet celebrity” and “pseudo-scientist” are the two most frequently used words. “Pseudo-scientist” is a term often used by netizens to ridicule those who claim to be experts who are not authorities. Here, some netizens used the term “pseudo-scientist” to describe the two experts, expressing their doubts about their authority. Some representative quotes include: “He is a pseudo-scientist, he is more like a businessman,” and “Why can such pseudo-scientists still come out and speak now?” Another term frequently used to describe experts is “internet celebrity”. “Internet celebrity” originally meant a person with a certain number of fans on the internet, which was a neutral expression. However, its use by anti-intellectuals to describe medical professionals is linked to the idea that they exist and that medical professionals appear frequently in the media because they want to increase their exposure rather than address COVID-19-related issues. As a result, netizens have labeled the experts as “internet celebrities” to express their discontent. This is reflected in statements such as “He does nothing but see (appears on) the media every day; he is an internet celebrity” and “Is there any hope in fighting the epidemic with the words of an internet celebrity?”.

In addition, anti-intellectuals often use terms such as “bragging”, “disaster to the country” and “lapdog”. “Bragging” is used to attack experts for being misinformed, which demonstrates the low level of public trust in experts. Examples include “This new drug is Zhang’s spiritual bragging and self-deception!” and “Bragging is popular now!” “Disaster to the country” is a negative label used by the audience to describe experts as a disaster for the country because they consider their advice to be useless or even harmful, to the detriment of the country. Some examples of statements are “It’s a disaster for the country to rely on such pseudo-scientists for epidemic prevention.” “Lapdog” is also a negative tag that is used to describe the experts as being instructed by others (stakeholder organizations or governments or potentially hostile forces in China). Some statements such as “Doctor Zhang is a lapdog of capitalism,” show this.

### 4.3. Specific Features of Anti-Intellectual Texts

By coding the collected texts, we summarized 10 characteristics shared by these texts: (a) claiming that the expert’s previous advice was invalid, (b) questioning the lack of evidence for the experts’ statements, (c) suspicion that the experts were motivated by profit, (d) the experts’ advice was in conflict with the commenter’s self-interest, (e) the experts’ advice failed to deliver immediate benefits, (f) conflict between the expert’s advice and personal emotions; (g) logical fallacies in own deductions, (h) the experts’ advice did not fit with established knowledge, (i) direct verbal abuse and (j) belief in rumors. A secondary analysis of these 10 characteristics to extract their commonalities revealed that they could be grouped into three categories: (a) questioning the authority of experts, (b) the pursuit of direct benefits and conflicts of interest and (c) irrational behavior. The specific situation is shown in Table 2.

## 5. Discussion

The analysis of the data revealed that anti-intellectualism in health information dissemination in the context of COVID-19 was not under-represented on internet platforms. The numbers of social media users who supported and opposed the experts and their advice were nearly identical. We also found that labeling is common in anti-intellectual texts, with anti-intellectualists tending to build up a perceived negative image of the experts through their own judgements of them. However, the existence of this phenomenon is as much related to the irrational behavior and instrumental rationality of society at large as it is to the decline in the authority of experts. Next, we analyzed the phenomena revealed through the study and sought strategies to effectively balance the relationship between anti-intellectualism and efficient health communication.

### 5.1. The Authority of Experts and Scholars Has Declined

In our findings, anti-intellectuals dismissed experts and their advice on the grounds that they questioned the authority of experts. This is related to anti-intellectuals’ perceptions of experts. Our study suggests that some anti-intellectuals questioned the advice of experts because of a perception that the experts were driven by profit and were endorsed by certain organizations. This finding has parallels with some previous studies that suggest that the public considers experts’ advice to not be purely out of consideration of the social effects but instead driven by certain forces [35,36]. Furthermore, this skepticism is also related to some of the behaviors of the expert scholars themselves. For example, the public’s trust in experts is reduced if their previous advice has been ineffective. If an expert is unable to provide solid evidence when stating an opinion or providing advice, this can lead to questioning by others. With regard to the ineffective advice of experts, some studies have argued that the failure of expert advice is inevitable, especially in new emergencies [56,57]. For example, in the face of the COVID-19 outbreak, experts needed time to analyze the situations before they could give appropriate advice. However, in a sudden emergency such as this, a panicky public wants to be informed quickly or else they see the experts as failing [58,59]. In such cases, the information the expert can give is not always accurate and often quickly becomes ineffective, leading the public to question the expert’s competence. Nowadays, audiences in an environment of massive information demand faster and faster dissemination of information, and the patience of audiences is gradually decreasing. As a result, if experts are unable to provide effective information quickly in an emergency, the public will question the competence of the experts and scholars. As a result, experts are caught in a situation where they need to provide effective information to the public but lack sufficient time to collect information and data for research.

Regarding the inability of experts to present solid evidence when giving advice, this is related to the length limits of posts on social media platforms. For example, Sina Weibo, the mainstream social media software in China, limits blog posts to 160 characters. This limits the behavior of experts when they communicate with the general public on social media platforms. Experts are often limited to stating their own conclusions without being able to fully present the sources of their data and the process of deriving them. Moreover, anti-intellectuals often lack the ability to think rationally [21], which leads them to oppose the advice of experts and even to attack expert scholars in an emotionally driven manner.

### 5.2. Unreflective Instrumentalism and the Pursuit of Self-Interest

In our research, we found that the public would oppose expert advice if it did not yield immediate benefits quickly enough. This is because they perceive it to be ineffective advice, even though these benefits can manifest themselves over time. Previous research has argued that unreflective instrumentalism that suppresses questions about the ends towards which practical and efficient means are directed is a form of anti-intellectualism [20,23]. Our research findings are consistent with previous research results. The contempt for theoretical knowledge is seen as a consequence of unreflective instrumentalism. For anti-intellectuals, experts and intellectuals are only concerned with theories that do not directly solve problems, and the suggestions they make lack feasibility. Some anti-intellectuals are even blindly opposed to data and experimental results that can be used as evidence because they believe that this knowledge is less valid than accumulated life experiences.

In addition, our study found that some anti-intellectuals resist expert advice when it conflicts with their personal interests, believing that the experts’ advice is unreasonable. This was reflected in Chinese COVID-19 prevention and control practices, where experts and scholars advised audiences on social media to wear masks and stay away from crowded places, and despite studies suggesting that these health measures can help individuals stay safe from COVID-19 [60,61], some members of the public resisted this advice because it restricts their freedom in the short run.

### 5.3. Specific Expressions of Irrational Behavior

Our findings identified five specific manifestations of irrational behavior: emotional dominance, abusive behavior, overconfidence and trusting rumors. Of these, overconfidence and emotional dominance have been reported in previous studies [26,27,28]. Previous studies suggest that those two features are the result of irrational anti-intellectuals being time-poor and eschewing rational logical thinking [21,25]. Overconfidence makes the anti-intellectual more confident in their personal experience and logically derived results, even though their experience and logic are not correct. When confronted with advice from experts, anti-intellectuals do not judge the credibility of information through searching for evidence and rational logical thinking, as they deny the validity of such thinking. This ultimately leads to a conflict between their ideas and the advice of the experts.

Emotional dominance is also a result of trivializing rational thinking, which suggests that audiences choose whether to accept expert advice based on emotional perceptions rather than the validity of the advice. On social media platforms, the rate of emotional contagion is increasing [62,63]. This means that some negative emotions can spread more quickly and social media users are more susceptible to the emotions of others. The spread of such sentiments can easily bring together emotionally driven anti-intellectuals and lead to irrational mass incidents against experts.

In addition, social media platforms also influence the level of trust in rumors among the users of social media platforms [64,65], thus indirectly contributing to the rebellion of some members of the public against experts and their advice. Trusting rumors is a potential factor that influences the public’s experience and shapes their emotions [66,67]. Overconfident anti-intellectuals tend to believe rumors more than the truth, which as a result is the antithesis of rumors. The spread of rumors on social platforms is therefore also linked to the emergence of the phenomenon of anti-intellectualism.

Abusive behavior was also demonstrated in the study results. On Chinese social media, a large number of social media users who oppose experts engage in abusive behavior towards them. Labeling tactics, i.e., applying a number of negative labels to experts, were a common part of the abusive behavior. Users have labelled experts as “internet celebrities”, “lapdogs” and so on. These labels reflect the anti-intellectuals’ perceptions of experts and simply reflect their reasons for opposing them. At the same time, such slogans are easily spread [68], thus accelerating the construction of a negative image of experts.

### 5.4. Balancing Anti-Intellectualism and Efficient Health Communication

Previous studies have shown that anti-intellectualism can have a negative impact on health information dissemination [12,13,14]. Nevertheless, anti-intellectualism cannot be completely eradicated. Because its existence prevents experts and intellectuals from becoming very powerful, and because anti-intellectualism is an attitude [18], it is unrealistic to eliminate it completely. Based on the characteristics of anti-intellectualism, we can make some suggestions in terms of the dimensions of the subjects involved.

Firstly, the public should regularly and actively educate themselves on media knowledge. Emotional contagion and the spread of rumors on social media platforms have influenced the public perception of information, thus contributing to the development of anti-intellectualism. In view of this, social media users should try to improve their discernment regarding information and media literacy to enhance public and social awareness on anti-intellectualism [69,70]. In addition, social media users need to improve their sense of social responsibility and discipline their own behavior. For example, they should be suspicious of information that they are not sure of and not engage in abusive comments about others.

Secondly, the new media platforms play doubles roles as media channels and gatekeepers of information dissemination. New media communication professionals have the primary responsibility and technical ability to control the flow of information and build a healthy network environment [71]. Therefore, we suggest that new media platform professionals increase the level of supervision and develop necessary procedures to discipline account hosts and individuals that spread inaccurate information and engage in abusive behavior. A healthy and positive environment needs to be created for experts to communicate with the public.

Finally, experts and academics can be innovative in their methods of communication. Studies have found that because of the limitations of social media platforms regarding the text length of content, experts can often only communicate their conclusions on social media platforms, without the support of actual evidence and the process of derivation. Furthermore, audiences have difficulty in understanding some of the more specialized knowledge, which can easily lead them to misunderstand the information conveyed by experts. Therefore, experts can deliver information in creative formats, such as in the form of videos or graphics, which would help to increase the authority of the information and build public confidence in that information.

## 6. Conclusions

This study explores the presence of anti-intellectualism in health communication on new media platforms in China to understand its manifestations and characteristics on new media platforms and to try to provide strategies that can balance anti-intellectualism with efficient health communication. We selected blog posts by two Chinese experts in epidemic prevention giving advice about preventing and controlling COVID-19 and selected the comments responding to the blog posts for analysis. This study found, through textual investigation, that anti-intellectualism persists in new media platforms mainly in the form of (a) questioning the authority of the experts; (b) the pursuit of direct benefits and conflicts of interest and (c) irrational behavior. Therefore, this study suggests that health communication authorities need to build up the public’s media literacy. This study also suggests that social media platforms should create a healthy and positive communication environment to reduce the uncertainty of health information dissemination. Finally, experts should be innovative in the way health information is communicated, taking the needs and characteristics of the audience into account. The findings of this study have several important practical implications for improving the efficiency of health communication via social media platforms in China. The strength of this study is that we examined the manifestations of anti-intellectual behavior rather than the attitudes or thoughts of anti-intellectuals, unlike previous studies. In addition, our findings can be used to further refine the theory of “anti-intellectualism”.

### Limitations of the Study

There are several limitations of this study. Firstly, regarding the sample size, we extracted 1540 tweets from two microblogs, both of which originated from two of the most followed medical experts in China, whose blogs revealed various opinions in the comments section. Although the selected sample is representative, a larger sample would have provided a more comprehensive and definitive picture of the characteristics and generative logic of communicating health information on China’s new media platforms.

Second, we mainly examined two cases of COVID-19 health information dissemination on online platforms; further studies could expand the discussions to other contexts of health information communication to validate our research findings. Last but not least, as our research findings are bounded by the social and cultural context of China, it would be meaningful to explore social contexts in other societies and regions to further develop the knowledge on anti-intellectualism.

## Figures and Tables

**Figure 1 healthcare-11-00121-f001:**
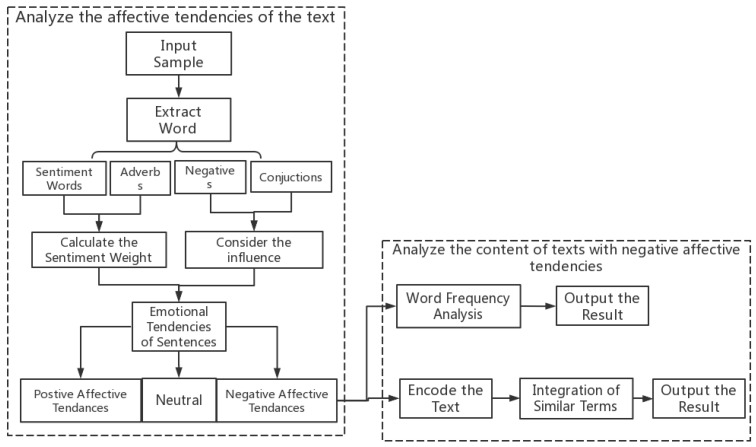
The framework of data analysis.

**Figure 2 healthcare-11-00121-f002:**
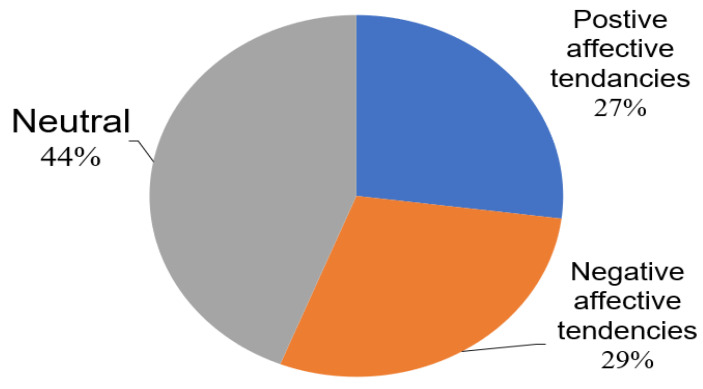
Analysis of the emotional tendencies of the keywords in the sample.

**Figure 3 healthcare-11-00121-f003:**
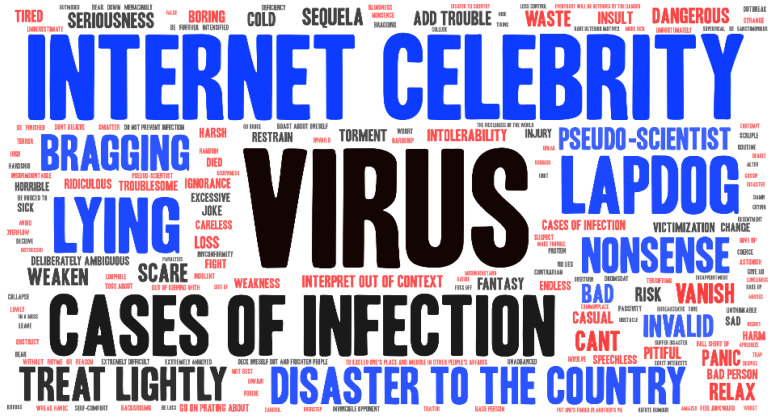
Keywords with negative affect.

**Table 1 healthcare-11-00121-t001:** Keyword frequency statistics (partial list).

Keyword	Word Frequency
Virus	69
Internet celebrity	29
Cases of infection	19
Treat lightly	16
Lapdog	14
Nonsense	11
Disaster to the country	10
Pseudo-scientist	9
Vanish	8
Lying	8
Bragging	8
Invalid	7
Bad	7
Weaken	6
Relax	6
Seriousness	5
Dangerous	5
Panic	5
Scare	5
…	…

**Table 2 healthcare-11-00121-t002:** Characteristics of anti-intellectual texts.

Category	First-Level Coding	Typical Text
Questioning the authority of experts	The expert’s previous advice was invalid	① The epidemic has been going on for two years now and Doctor Zhang is always fond of making unrealistic statements and constantly breaking his word! ② Please take responsibility, internet celebrity pseudo-scientist! I don’t want to see your word being broken after a period of time!
	Questioning the lack of evidence for experts’ statements	① There are no data to support and justify this conclusion! ② No official data have been released; did you make them up yourself? ③ This is your bright idea, right?
	Suspicion that the experts are motivated by profit	① You are the lapdog of capitalism. ② Colluding with capitalism, brutalizing the people, calling yourself a national scholar but you are actually a national traitor!
Unreflective instrumentalism	Expert advice in conflict with self-interest	① We’re not wallpaper; dynamic prevention and control is unacceptable! ② Your advice has made my life extremely inconvenient!
	Expert advice fails to deliver immediate benefits	① If your advice is valid, why is the virus still widespread? ② Your advice is no quick fix for the outbreak!
Irrational behavior	Conflict between expert advice and personal emotions	① Why should Chinese measures be copied from abroad? ② Domestic vaccines are the best!
	Logical fallacies in their own deduction	① (Vaccine) does not protect against minor illnesses but instead against major illnesses? Ridiculous logic! ② Vaccines are useless, as is wearing a mask; otherwise, there would be no need for isolation.
	Expert advice does not fit with established knowledge	① Whether it’s serious or not has to do with physical fitness and has nothing to do with vaccines; you need to stop lying! ② Atypical pneumonia is automatic and has nothing to do with vaccines, which are useless against these epidemics.
	Direct verbal abuse	① You’re a fucking liar. ② You don’t deserve to be called a scientist; you’re an internet celebrity!
	Believing rumors	① He even faked his PhD thesis! [This has been proven otherwise] ② He took money from vaccine companies to promote their products. [No evidence]

## Data Availability

The data presented in this study are available on request.

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
