# Peer review of "COVID-19, Anti-Intellectualism, and Health Communication: Assessing the Chinese Social Media Platform Sina Weibo"

_healthcare, 2022, doi:10.3390/healthcare11010121_

Round 1

Reviewer 1 Report

The topic is interesting and novel in its analysis, however the sample is scarce as it only sticks to two scientific articles to analyze their comments.

The theoretical framework needs to delve deeper into its descriptors to give depth to the subject of analysis that in other countries can be called negationism.

Basically the topic is important, there is little scientific literature on the matter. It would be good to know how to deal with these social movements based on denialism.

The sample should increase or for example: include two articles for China, two for Europe, two for Asia, two for America and two for Australia.

Author Response

Comment 1: The topic is interesting and novel in its analysis, however the sample is scarce as it only sticks to two scientific articles to analyze their comments.

Author response: Though our data is based on two micro-blogs, having 1540 tweets. We also put the response of a reviewer in our study limitation part on pages 13 and 14.

Author action: We updated the manuscript by making appropriate changes (pp. 13-14).

Comment 2: The theoretical framework needs to delve deeper into its descriptors to give depth to the subject of analysis that in other countries can be called negationism.

Author response: We have added few sentences to address the concerns of reviewer under the title of “Theoretical framework”.

Author action: We updated the manuscript on pages 3-6.

Comment 3: Basically the topic is important, there is little scientific literature on the matter. It would be good to know how to deal with these social movements based on denialism.

Author response: We do value to reviewer comments and suggestion. We believe it’s a good insight to work on “dealing with these social movements based on denialism”.

Author action: The reviewer has highlighted a very important issue. We may take this issue in our future research work. We are very thankful for the reviewer’s insightful suggestions.

Comment 4: The sample should increase or for example: include two articles for China, two for Europe, two for Asia, two for America and two for Australia.

Author response: We do value to reviewer comments and suggestion. Though, at this stage, we believe it’s hard to add the sample size. We may consider this suggestion in our future studies.

Author action: At this stage we are afraid to add a new sample as it will change the entire study.

Reviewer 2 Report

Please complete the purpose and hypotheses/research questions in the paper, and at the end add the strengths (there are a lot of them!) and limitations of the research conducted. It is hard for me to comment on the methodology, as the study is quite unusual, but very interesting. Please standardize the spelling of the name COVID-19 (like this).

Author Response

Comment 1: Please complete the purpose and hypotheses/research questions in the paper

Author response: We have added few sentences to address the concerns of reviewer under the title of “Theoretical framework”.

Author action: We updated the manuscript on pages 3-6.

Comment 2: At the end add the strengths (there are a lot of them!) and limitations of the research conducted.

Author response: We have addressed the reviewer’s concerns on pages 13 and 14 by adding the strengths and limitations of the study.

Author action: We updated the manuscript by making appropriate changes (pp. 13-14).

Comment 3: Please standardize the spelling of the name COVID-19 (like this).

Author response: Thank you.  The manuscript has been subjected to rigorous editing.

Author action: Yes, errors corrected through “a careful proofreading.” We hired the services of MDPI English language editing. The certificate of English editing is attached too.

Reviewer 3 Report

Comments It is an interesting research topic. The paper could be improved. In the introduction, the authors has described the research background [till line 68], it is suggested that they can clearly illustrate the research gap, the research question, and theoretical contribution of the research. The introduction section can be concluded by a brief description of the structure of the paper. In the second section of the theoretical framework, anti-intellectualism were defined and the features of anti-intellectualism in health communication was described and discussed. At the end of this section, the problems in the research domain of anti-intellectualism in health communication were identified: (1) limited number of studies, lack of empirical studies; (2) no strategies related to anti-intellectualism and health communication. These statements need to be further discussed and substantiated with evidence from the existing literature. The following questions can be explored: what research has been conducted; what has not been studied; what research methodology/method has been employed. Based on this, the potential research gap and research questions can be identified and justified. However, the research question proposed by the authors as “Therefore, this paper will examine the performance of anti-intellectualism on Chinese Internet platforms through empirical analysis and propose corresponding balancing measures”. The research question was not clear and the the theoretical contribution of the research needs to be articulated and justified. In the section of research methodology, you did not describe the research methodology of the research. You started from sampling. Then you described the textual analysis. And the process of the research. In this section, you need to clearly describe the research question, research objectives and then justify your research methodology. Then you can describe the method you used and the process of the research. and how the data were analysed. In the section of the results, you identified some themes from the data. In the section of discussion, you discussed many themes. You needed to make more connections between what you found from your empirical research and the extant literature. Need some proof reading, some sentences were not well-written and difficult to understand. Thanks.

Author Response

Comment 1: In the introduction, the authors has described the research background [till line 68], it is suggested that they can clearly illustrate the research gap, the research question, and theoretical contribution of the research.

At the end of this section, the problems in the research domain of anti-intellectualism in health communication were identified: (1) limited number of studies, lack of empirical studies; (2) no strategies related to anti-intellectualism and health communication. These statements need to be further discussed and substantiated with evidence from the existing literature.

Author response: We have addressed the concerns of reviewer under the title of “Theoretical framework”.

Author action: We updated the manuscript on pages 3-6.

Comment 2:  The introduction section can be concluded by a brief description of the structure of the paper.

Author response: We have added few sentences to address the concerns of reviewer on page 2.

Author action: We updated the manuscript on page 2.

Comment 3:  In the section of research methodology, you did not describe the research methodology of the research. You started from sampling. Then you described the textual analysis. And the process of the research. In this section, you need to clearly describe the research question, research objectives and then justify your research methodology.

Author response: We have added a paragraph to address the concerns of a reviewer on pages 5 and 8.

Author action: We updated the manuscript on pages 5 and 8.

Comment 4:  In the section of the results, you identified some themes from the data. In the section of discussion, you discussed many themes. You needed to make more connections between what you found from your empirical research and the extant literature.

Author response: We have addressed the concerns of a reviewer on pages 12 and 13.

Author action: We updated the manuscript on pages 12 and 13.

Comment 5:  Need some proof reading, some sentences were not well-written and difficult to understand.

Author response: Thank you.  The manuscript has been subjected to rigorous editing.

Author action: Yes, errors corrected through “a careful proofreading.” We hired the services of MDPI English language editing. The certificate of English editing is attached too.

Round 2

Reviewer 3 Report

The issues I raised have been addressed. I think the paper has greatly improved. However, the authors need to be careful about what they claimed and what was delivered. For example, in the abstract, it says that  “against this backdrop, this study has applied textual analysis to explore the key factors that have contributed to the development of anti-intellectualism” (line 15, p.24). I cannot see this issue has been addressed in the paper. 

Author Response

We have addressed the concern of the reviewer in the abstract, lines 7-8. We are thankful to the reviewer for highlighting the issue.